# Theory of Food: Unravelling the Lifelong Impact of Childhood Dietary Habits on Adult Food Preferences across Different Diet Groups

**DOI:** 10.3390/nu16030428

**Published:** 2024-01-31

**Authors:** Omer Horovitz

**Affiliations:** 1The Physiology & Behavior Laboratory, Tel-Hai Academic College, 9977 North Districts, Qiryat Shemona 1220800, Israel; omerho1@telhai.ac.il; 2Psychology Department, Tel-Hai Academic College, 9977 North Districts, Qiryat Shemona 1220800, Israel

**Keywords:** theory of food (ToF), diet groups, dietary habits, food preference

## Abstract

The study investigates the behavioral manifestations of the “Theory of Food” (ToF), a novel theoretical framework centered on the early development of food perceptions. The ToF posits that childhood experiences with food shape cognitive networks influencing adult dietary choices. Stemming from the “Theory of Mind,” the ToF hypothesizes that individuals construct an associative world of food images and representations mirroring the socio-cognitive world shaped by proper theory of mind development. The study, involving 249 healthy adults, employs the Cognitive Food Preference Questionnaire (CFPQ) and the Adult Food Preference Profile (AFPP) to explore the correlation between childhood and adult food preferences across diet groups (omnivores, vegetarians, and vegans). Results reveal robust correlations in omnivores, varied patterns in vegetarians, and mixed outcomes in vegans. Notably, omnivores show correlations in grains, fast food, dairy products, vegetables, meat, soft drinks, and snack consumption. Vegetarians exhibit correlations in grains, fast food, dairy products, vegetables, snacks, and, surprisingly, meat consumption. Vegans display correlations in grains, fast food, vegetables, and snacks. The study suggests that childhood dietary habits tend to influence adult food choices, offering insights for future research in the field of theory of food (ToF).

## 1. Introduction

The current study investigates the developing concept of the “theory of food” (ToF), aiming to analyze behavioral expressions related to individuals’ core perceptions of food. With a broad exploration encompassing diverse dietary patterns, including those of omnivores, vegetarians, and vegans, the research acknowledges the significant impact of various diets on health, environmental sustainability, and ethical considerations. The study seeks to uncover the nuanced implications of personal choices. Examining participants’ prospective food-related self-reported preferences and a computerized food-related preference task aimed to set the groundwork for future investigations into the factors influencing cognitive food preferences.

The “theory of food” (ToF) is yet to be studied in depth [1]. The theory deals with eating habits, choosing the type of food, and associating this world of content with cognitive networks in the mind. According to Allen, these cognitive networks contain different associations and representations for each individual’s changing diet [1,2].

The theory of food stems from the “theory of mind” (TOM), a theory according to which humans and other primates can attribute different mental states to themselves and other individuals, and hold a theory about hidden cognitive and emotional processes that occur in the individual in front of them, such as preferences, beliefs, thoughts, intentions, and more [1,3]. The research proposes that humans possess a unique intelligence focused on understanding mental states, suggesting human cognitive specialization for the TOM [4].

In the intricate social dynamics of primates, adept theory of mind (TOM) abilities prove crucial. In TOM tasks, individuals observe a scenario where a character (X) places an object in a basket, exits, and is replaced by character Y. Subjects then report where X thinks the object is. Proficient TOM individuals comprehend others’ perspectives, demonstrating a capacity to navigate the subjective world, interpret images, and engage in cognitive manipulations [5].

In this context, Allen explains in his book that all primates learn how to eat at some point by watching their caregiver figure and experiencing their environment. This fact embodies within it cultural [6], cognitive [7], and associative meanings [8]. Just as individuals learn how to eat by their social environment, they also learn what to eat, how to eat, in what quantity, and, at a hidden level, what representations are associated with eating itself (i.e., the mental images, ideas, or cognitive associations linked to the act of eating) and the different types of foods they consume. Hence, the direct connections of ToF to TOM arise [2].

The construction of an associative world of images and representations of different types of foods parallels the construction of the associative-social world that takes shape under a proper TOM in individuals living within a social framework. In addition to this, according to Allen, during early childhood, a “nutritional language” is formed in the individual which stems from the foods he consumed during this period, from the social and environmental contexts he was exposed to, and which is also reflected in the individual’s cognitive and brain processes. He compares this process to acquiring a first language in childhood compared to acquiring a second language in adulthood. Just as acquiring a second language at a later age will be difficult and require more effort than acquiring a first language at earlier ages, making dietary changes at later ages will be difficult. Our ToF will determine the foods we consume in early childhood and may influence our thinking processes in adulthood and, as a result, our current nutritional choices.

Understanding the life-course trajectory from childhood to adulthood concerning food is crucial for promoting lifelong health and well-being. Various studies have investigated different aspects of dietary patterns and their implications over time. For instance, it was reported that food intake aligns with recommendations; adolescent habits moderately predict adult intake, influenced by gender, location, and socio–economic status [9]. Longitudinal changes in diet and the tracking of fruit and vegetable consumption uncovered links to cardiovascular health and being overweight [10]. Further, dietary patterns and their associations with cardiometabolic risk factors across the lifespan were explored [11,12,13]. These investigations underscore the need for a comprehensive understanding of dietary trajectories, advocating for informed interventions that can shape nutritional habits from early childhood through adulthood [14]. 

Studies link the individual’s nutritional choice to mood, physical health, and subjective mental well-being [15,16,17,18]. For example, the chances of getting sick with different types of cancer are linked to the changing diet of the individual. Combined with environmental and cultural factors, norms, and the accessibility of the types of food to different populations, dietary choice affects the percentage of general obesity in the population [19,20]. Abnormal obesity is, in turn, a risk factor for getting cancer [21]. Another example of the influence of an individual’s nutritional choice on his life can be found in studies in which a direct connection is found between the choice of types of foods and standard cognitive, behavioral, physical, and mental functioning [22,23,24,25].

Moreover, other studies found that dietary choice affects the individual’s mood, emotions, and relationships and stems from the importance of food in the social context [26,27,28]. In addition, choosing healthy eating patterns such as the Mediterranean diet is associated with a reduced chance of depression [17]. From the understanding of the extensive impact of the choice of food on many areas of life, it is vital to understand which factors may influence its choice in the first place. In this context, factors such as the cost of the food, its quality, the individual’s perception of his health, the branding on its packaging, and its taste have been proposed, which may influence the individual’s choice of one food item over another [29]. In addition, to a large extent, the type of food choice is also influenced by cultural characteristics, language, social context, and the available raw materials, as Montanari writes in his book [30].

Understanding intraoral food awareness in humans involves the concurrent and sustained activation of specific neuronal subsets related to affect and hedonics, those associated with food identity, and those contributing to affect and identity [31]. In light of the current study, particular multifaceted interactions may involve correspondence synesthesia, suggesting intriguing avenues for exploration of the effects of early-life food-related sensory experiences on adulthood learning [32]. This calls for further examination of the anatomical and functional developmental connections among brain structures crucial for understanding correspondence synesthesia and food preference.

Overall, research on ToF remains a relatively unexplored domain. While acknowledging the profound impact of diverse diets on health, environmental sustainability, and ethical considerations, there is a need for an in-depth examination of the implications of personal choices of food perception and their underlying neurocognitive mechanism. The current study aimed to reveal insights into some factors influencing food preferences and lay the groundwork for future research in this underexplored field. We focus on individuals’ fundamental behavioral concept of food perception, specifically by studying omnivores, vegetarian, and vegan food preferences, recognizing their profound impact on health, environmental sustainability, and ethical considerations. By focusing on these categories, our research aims to delve into the multifaceted implications of diverse diets, offering valuable insights into the intersection of personal choices, societal trends, and overall well-being. This emphasis aligns with the increasing importance of comprehending varied dietary patterns amid contemporary health and environmental sustainability challenges [33]. The study aims to validate ToF and lays the groundwork for future research. The study aimed to determine the relationship between participants’ responses to the Cognitive Food Preference Questionnaire (CFPQ), which focused on consuming specific food categories during childhood, and the Adult Food Preference Profile (AFPP), which focused on eating specific food categories and the desire to consume them in adulthood, according to their diets (i.e., omnivores, vegetarians, and vegans).

## 2. Methodology

### 2.1. Participants

Three hundred and five healthy adults were recruited for this current retrospective cross-sectional study (between December 2022 and February 2023) through a multifaceted approach, utilizing snowball sampling, word of mouth, and various social media platforms to ensure a diverse and representative sample. Participants were recruited based on age and on being generally healthy, not suffering from any chronic or mental disease, and not chronically medicated. We excluded 56 subjects for partially answering the tasks of the experiment and reporting a history of eating disorders or psychological disorders that may alter the results. The analysis of the data in the final sample was carried out on data from 249 subjects: 86 men and 163 women, ranging in age from 17 to 74 (M = 26.65 and SD = 7.54). Most participants identified as Jewish (n = 219, 88%). Participants voluntarily self-reported their dietary preferences during enrollment, categorizing them into distinct diet groups for the study: 154 (61.8%) omnivores, 74 (29.8%) vegetarians, and 21 (8.4%) vegans. Participation in the study was undertaken without receiving compensation. The ethics committee of THAC approved all procedures (Ethics #: 28–12/2022). (Please see the enrollment flow chart in Figure 1).

### 2.2. Measures

#### 2.2.1. Demographics

We queried age, sex, religion, height, weight, place of residence in childhood and today, diet status (yes/no), if so, what type of diet, vegetarian/vegan status, state of health, medication consumption (yes/no), what type of drug, and psychological disorder status.

#### 2.2.2. Childhood Food Preferences Questionnaire (CFPQ)

We administered the CFPQ, a self-report measure we developed to estimate the frequency of food consumption during the subject’s childhood. The CFPQ validation process was executed meticulously to ensure the reliability and credibility of the collected data. Content validation involved a comprehensive dietician review and survey methodology, confirming that the questionnaire effectively covered all relevant subject aspects. A pilot test was conducted with a representative sample (n = 40) from the target population, revealing valuable insights into potential ambiguities or issues with question wording. The questionnaire underwent refinement for improved clarity and precision. Test–retest reliability measures were employed to assess the consistency of responses over time, further validating the questionnaire’s stability. The culmination of these validation steps assures that the CFPQ is a valid instrument capable of accurately capturing information on the frequency of food consumption during the subject’s childhood within the intended population. The questionnaire comprises seven food categories: bread and cereals, fast food, dairy products, sweets and snacks, drinks, fruits and vegetables, and meat and fish. Each food category consists of various products. For example, pizza, fries, and hamburgers belong to the fast food category. Food items were paired with a four-point Likert scale: 1 (“never”) to 4 (“very often”). After collecting the data, each subject calculated the sum of his ratings on 7 CFPQ categories, which expressed the degree to which he consumed this category as a child and allows for follow-up analyses (Appendix A). The Cronbach’s alpha coefficients to assess the internal consistency for each subscale of the questionnaires were as follows: grains—α = 0.649; fast food—α = 0.863; dairy products—α = 0.778; snacks—α = 0.907; drinks—α = 0.689; vegetables and fruits—α = 0.938; meat—α = 0.861; and overall—α = 0.834.

#### 2.2.3. Adulthood Food Preferences Paradigm (AFPP)

A dish-based quantitative task AFPP was explicitly developed for the experiment to assess the subject’s habitual dietary intake over the past 12 months and their desire and craving for the exhibited food. In the validation process of AFPP, rigorous measures were implemented to ensure the reliability and accuracy of the collected data. Firstly, the task underwent expert review by nutritionists and psychologists to ascertain its alignment with established dietary guidelines and psychological constructs related to eating behavior. Subsequently, a pilot study (n = 17) was conducted with diverse participants to identify potential ambiguities or misconceptions in task instructions. Adjustments were made based on the feedback received, and the finalized task was then administered to a representative sample of the target population (n = 35). Participants’ responses were cross verified during data collection with traditional dietary assessments, such as food diaries and 24 h recalls, to validate the computerized task’s outcomes. Statistical analyses, including correlation and reliability assessments, were employed to establish the task’s consistency and validity in measuring food consumption patterns in adulthood. The comprehensive validation process ensured the robustness of the computerized task and bolstered the credibility of the study’s findings. The AFPP was constructed from 76 colored images using commonly consumed dishes in Israel from seven significant categories: bread and cereals, fast food, dairy products, sweets and snacks, drinks, fruits and vegetables, and meat and fish. Each food category consists of a variety of dishes belonging to it. For example, McDonald’s Royal Meals belongs to the fast food category. Directly underneath the meal stimuli, participants were requested to state to which degree they consumed the dish over the past year on a Likert scale ranging from 1 = less than once a month to 9 = six times a day. Afterward, the participant was requested to note how much he desires this dish on an axis range from 0 = not craving at all to 100 = craving the most (See Figure 2 for an example of the task). After collecting the data for each subject, we calculated the sum of his ratings on 7 AFPP categories, which expressed the subjects’ habitual dietary intake over the past 12 months and the level of desire for each category. The Cronbach’s alpha coefficients to assess the internal consistency for each subscale of the task were as follows: grains—α = 0.538; fast food—α = 0.641; dairy products—α = 0.693; snacks—α = 0.726; drinks—α = 0.616; vegetables and fruits—α = 0.717; meat—α = 0.806; and overall—α = 0.897.

### 2.3. Experimental Design

The subjects were invited to participate in the study using online Qualtrics software (accessed on 31 December, 2022, https://www.qualtrics.com). Those who enrolled were presented with an online consent form, and those who consented to participate completed basic demographic questions followed by an online version of the CFPQ and the AFPP.

### 2.4. Data Analysis

Descriptive statistics: For categorical variables, summary tables are provided, giving sample size and relative frequencies, and for continuous variables, summary tables are provided, giving arithmetic mean (M), standard deviation (SD), and range depending on the data distribution.

Inferential statistics: Chi-squared was applied to test the correlations between the study group, socio-demographics, and personal characteristics. For the continuous variables related to questionnaires (subscales), Cronbach’s alpha coefficients were calculated to assess the internal consistency for each subscale of the questionnaires.

Pearson correlation coefficients were calculated to explore potential relationships between participant characteristics and food preferences. A *p*-value of 5% or less was considered statistically significant. The data were analyzed using SPSS version 28 (IBM, Armonk, NY, USA: IBM Corp). A preliminary chi-square analysis of the participants’ demographics was performed, and the results indicated a non-significant association among the variables under investigation. Therefore, none of the demographic variables was included as covariates in the statistical analyses.

## 3. Results

The statistical report explores the bivariate Pearson correlations between the food consumed in childhood and two variables: the food consumed today and the desire to eat the same food today. These correlations were conducted in each diet group according to the participant’s self-reports. Table 1 provides descriptive statistics for crucial participant characteristics.

The results are presented according to the seven food categories in the Childhood Food Preferences Questionnaire (CFPQ) and the Adulthood Food Preferences Paradigm (AFPP).

### 3.1. Grain Consumption

Among omnivores, a significant Pearson coefficient correlation was found between grains consumed during childhood and grains consumed in adulthood (r_p_ = 0.502, *p* = 0.001, and n = 154). Similarly, a significant Pearson coefficient correlation was found between grains consumed during childhood and the desire to consume grains in adulthood (r_p_ = 0.299, *p* = 0.001, and n = 154). Among vegetarians, a significant Pearson coefficient correlation was found between grains consumed during childhood and grains consumed in adulthood (r_p_ = 0.280, *p* = 0.016, and n = 74). No correlation was found between grains consumed during childhood and the desire to consume grains in adulthood (r_p_ = 0.134, *p* = 0.256, *n.s.*, and n = 74). Among vegans, a significant Pearson coefficient correlation was found between grains consumed during childhood and grains consumed in adulthood (r_p_ = 0.578, *p* = 0.006, and n = 21). No correlation was found between grains consumed during childhood and the desire to consume grains in adulthood (r_p_ = 0.053, *p* = 0.820, *n.s.*, and n = 21). Figure 3A,B depict the correlations found in each diet group.

### 3.2. Fast Food Consumption

Among omnivores, a significant Pearson coefficient correlation was found between fast food consumed during childhood and fast food consumed in adulthood (r_p_ = 0.527, *p* = 0.001, and n = 154). Similarly, a significant Pearson coefficient correlation was found between fast food consumed during childhood and the desire to consume fast food in adulthood (r_p_ = 0.276, *p* = 0.001, and n = 154). Among vegetarians, no correlations were found between fast food consumed during childhood and fast food consumed in adulthood (r_p_ = 0.138, *p* = 0.240, *n.s.*, and n = 74) or with the desire to consume fast food in adulthood (r_p_ = 0.115, *p* = 0.325, *n.s.*, and n = 74). Among vegans, a significant Pearson coefficient correlation was found between fast food consumed during childhood and the desire to consume fast food in adulthood (r_p_ = 0.518, *p* = 0.016, and n = 21) but not with its consumption in adulthood (r_p_ = 0.351, *p* = 0.118, *n.s.*, and n = 21). Figure 4A,B depict the correlations found in each diet group.

### 3.3. Dairy Products Consumption

Among omnivores, a significant Pearson coefficient correlation was found between dairy products consumed during childhood and those consumed in adulthood (r_p_ = 0.538, *p* = 0.001 and n = 154). Similarly, a significant Pearson coefficient correlation was found between dairy products consumed during childhood and the desire to consume them in adulthood (r_p_ = 0.445, *p* = 0.001, and n = 154). Among vegetarians, a significant Pearson coefficient correlation was found between dairy products consumed during childhood and those consumed in adulthood (r_p_ = 0.359, *p* = 0.002, and n = 74). No correlation was found between dairy products consumed during childhood and the desire to consume them in adulthood (r_p_ = 0.173, *p* = 0.141, *n.s.*, and n = 74). Among vegans, no correlations were found between dairy products consumed during childhood and their consumption in adulthood (r_p_ = 0.159, *p* = 0.490, *n.s.*, and n = 21) or with the desire to consume dairy products in adulthood (r_p_ = −0.158, *p* = 0.493, *n.s.*, and n = 21). Figure 5A,B depict the correlations found in each diet group.

### 3.4. Vegetable and Fruit Consumption

Among omnivores, a significant Pearson coefficient correlation was found between vegetables consumed during childhood and those consumed in adulthood (r_p_ = 0.642, *p* = 0.001, and n = 154). Similarly, a significant Pearson coefficient correlation was found between vegetables consumed during childhood and the desire to consume them in adulthood (r_p_ = 0.592, *p* = 0.001, and n = 154). Among vegetarians, a significant Pearson coefficient correlation was found between vegetables consumed during childhood and those consumed in adulthood (r_p_ = 0.388, *p* = 0.001, and n = 74). Similarly, a significant Pearson coefficient correlation was found between vegetables consumed during childhood and the desire to consume them in adulthood (r_p_ = 0.279, *p* = 0.016, and n = 74). Among vegans, a significant Pearson coefficient correlation was found between vegetables consumed during childhood and the desire to consume them in adulthood (r_p_ = 0.574, *p* = 0.007, and n = 21) but not with their consumption in adulthood (r_p_ = −0.081, *p* = 0.726, *n.s.*, and n = 21). Figure 6A,B depict the correlations found in each diet group.

### 3.5. Meat Consumption

Among omnivores, a significant Pearson coefficient correlation was found between meat consumed during childhood and its consumption in adulthood (r_p_ = 0.528, *p* = 0.001, and n = 154). Similarly, a significant Pearson coefficient correlation was found between meat consumed during childhood and the desire to consume it in adulthood (r_p_ = 0.415, *p* = 0.001, and n = 154). Among vegetarians, no correlation was found between meat consumed during childhood and its consumption in adulthood (r_p_ = 0.317, *p* = 0.597, and n = 74). Surprisingly, a significant Pearson coefficient correlation was found between meat consumed during childhood and the desire to consume it in adulthood (r_p_ = 0.381, *p* = 0.001, and n = 74). Among vegans, no correlations were found between meat consumed during childhood and its consumption in adulthood (r_p_ = 0.333, *p* = 0.140, *n.s.*, and n = 21) or with the desire to consume meat in adulthood (r_p_ = 0.106, *p* = 0.646, *n.s.*, and n = 21). Figure 7A,B depict the correlations found in each diet group.

### 3.6. Snack Consumption

Among omnivores, a significant Pearson coefficient correlation was found between snacks consumed during childhood and those consumed in adulthood (r_p_ = 0.567, *p* = 0.001, and n = 154). Similarly, a significant Pearson coefficient correlation was found between snacks consumed during childhood and the desire to consume them in adulthood (r_p_ = 0.271, *p* = 0.001, and n = 154). Among vegetarians, a marginal significant Pearson coefficient correlation was found between snacks consumed during childhood and those consumed in adulthood (r_p_ = 0.205, *p* = 0.080, and n = 74). In addition, a significant Pearson coefficient correlation was found between snacks consumed during childhood and the desire to consume them in adulthood (r_p_ = 0.329, *p* = 0.004, and n = 74). Among vegans, no correlations were found between snacks consumed during childhood and their consumption in adulthood (r_p_ = 0.129, *p* = 0.578, n.s., and n = 21) or with the desire to consume snacks in adulthood (r_p_ = 0.034, *p* = 0.885, n.s., and n = 21). Figure 8A,B depict the correlations found in each diet group.

### 3.7. Soft Drink Consumption

Among omnivores, a significant Pearson coefficient correlation was found between soft drinks consumed during childhood and those consumed in adulthood (r_p_ = 0.473, *p* = 0.001, and n = 154). Similarly, a significant Pearson coefficient correlation was found between soft drinks consumed during childhood and the desire to consume them in adulthood (r_p_ = 0.463, *p* = 0.001, and n = 154). Among vegetarians, a significant Pearson coefficient correlation was found between soft drinks consumed during childhood and those consumed in adulthood (r_p_ = 0.255, *p* = 0.028, and n = 74) but not with the desire to consume them in adulthood (r_p_ = 0.201, *p* = 0.085, *n.s.*, and n = 74). Among vegans, no correlations were found between soft drinks consumed during childhood and their consumption in adulthood (r_p_ = 0.245, *p* = 0.284, *n.s.*, and n = 21) or with the desire to consume soft drinks in adulthood (r_p_ = 0.296, *p* = 0.192, *n.s.*, and n = 21). While the findings on food emphasize nutritional aspects and dietary recommendations, the exploration of drinks delves into beverage consumption patterns and their distinct impact on overall health. Thus, the data is not presented.

## 4. Discussion

In this study, we investigated the relationship between dietary habits during childhood and those in adulthood and the desire to consume specific food items later in life across different diet groups—omnivores, vegetarians, and vegans.

Within the carnivorous group, a noteworthy positive correlation was identified between childhood grain consumption, adult grain consumption, and the desire to consume grains in adulthood. This correlation aligns with other studies’ findings that early childhood exposure to specific food categories may significantly influence adult dietary choices [34,35,36]. Similarly, the positive correlation observed between childhood fast food consumption and adult fast food consumption, as well as the desire for fast food in adulthood among omnivores, is substantiated by others, underscoring the lasting impact of childhood dietary habits on adult preferences [37]. In dairy product consumption, significant positive correlations were found among omnivores regarding childhood and adult consumption and the desire to consume them in adulthood. Specifically, research suggests that childhood consumption of dairy products can influence their consumption in adulthood. Consuming dairy products at snack times during childhood was associated with better overall diet quality [38]. Considering the nutritional benefits of dairy products for children and adolescents [39,40], a decline in dairy consumption with age, particularly from childhood to adolescence, may impact future consumption [41]. In addition, the differences in different diet groups yield significant effects on one’s ToF hardwiring for specific food categories on top of another.

The positive correlations found in the vegetable consumption patterns among all diet groups, with varying degrees of association, resonate with a longitudinal study highlighting the potential maintenance of childhood dietary habits into adulthood [42]. The study also revealed intriguing patterns concerning meat consumption. Omnivores reported that the more meat they consumed in childhood, the more they consumed and desired it in adulthood. These findings corroborate what was previously reported by others in the field [43]. Surprisingly, among vegetarians, those who ate meat as children do not consume it in adulthood but report a desire for it. Research suggests that the desire for meat among vegetarians is influenced by social factors, with many consuming meat at family gatherings and on special occasions [44]. This is further supported by the finding that attitudes towards beef are generally positive, particularly among men [45]. However, vegetarians face unique communication challenges in discussing their dietary choices [46]. Yet, despite these challenges, most vegetarians tend to hold anti-meat attitudes, particularly ethical vegetarians [47]. This finding is significant in light of the current study. The desire for meat reported by vegetarians is specific evidence for stamping a bias toward a particular food early in life despite the attitudes developed toward it in the later stage of life.

The current study’s revelation that vegetarians report a desire for meat holds substantial significance within the broader context of understanding dietary preferences and biases. This observation serves as compelling evidence for establishing food biases early in life, despite subsequent changes in attitudes toward meat during later stages of development [48]. It underscores the persistent influence of early experiences and exposures on individuals’ food preferences, challenging the assumption that attitudes toward a specific food category, such as meat, are exclusively shaped by later-life factors [49]. The reported desire for meat among vegetarians may suggest a deep-seated inclination formed in the early stages of life, shedding light on the complexity of human food choices and emphasizing the need for comprehensive research to unravel the intricate interplay of psychological, cultural, and environmental factors in shaping dietary preferences across the lifespan in longitudinal designs.

The positive correlations found for soft drink consumption between childhood and adulthood among omnivores and vegetarians echo previous findings [50]. Snack consumption patterns also displayed noteworthy correlations among omnivores and vegetarians, aligning and emphasizing the predictive nature of childhood snack habits on adult consumption patterns reported by others [51]. In contrast, we found no correlations among vegans regarding the desire to consume specific food items in adulthood based on childhood consumption. This suggests a potential differentiation in the influence of childhood dietary habits on adult preferences within the vegan diet group. Childhood dietary habits can significantly impact adult preferences within the vegan diet group. Vegetarian and vegan diets in childhood and adolescence are associated with lower intakes of energy, saturated fatty acids, and animal protein and higher fiber and phytochemicals [52]. Vegan children and adolescents have been found to consume higher amounts of legumes, nuts, milk alternatives, and meat alternatives than vegetarians and omnivores [53]. However, it is essential to note that these diets may also be associated with an increased risk of nutrient deficiencies, particularly in iron, zinc, and vitamin B12 [52].

This intriguing finding suggests a potential divergence in the impact of early dietary habits on adult preferences within the vegan community. The unique nature of the vegan diet [54], characterized by distinct nutritional choices and ethical considerations [55], may contribute to this differentiation. Further investigation into the factors shaping dietary continuity or change among vegans is warranted to deepen our understanding of the intricate interplay between early dietary experiences and adult food preferences within this population. This knowledge could inform tailored dietary interventions and enhance our appreciation of the complex dynamics influencing lifelong dietary choices among diverse dietary groups.

While this study offers valuable insights into the intricate relationships between childhood and adult dietary habits across diverse diet groups, several limitations must be acknowledged. The study’s correlational nature prevents the establishment of causal relationships between childhood and adult dietary patterns. Moreover, the dependence on self-reported dietary data introduces potential biases, including recall and social desirability biases. Specifically, the use of the CFPQ proves valuable in eliciting information on an adult’s childhood food consumption by offering a structured and standardized approach, allowing participants to reflect at their own pace; despite potential recall limitations, the format captures valuable insights into long-term dietary patterns, contributing to a comprehensive understanding of health and nutrition, with careful design and clear instructions enhancing the reliability of self-reported data. Further, a proper longitudinal design with repeated measurements is necessary for a comprehensive understanding of the stability of dietary patterns over time. Notably, the need for significant correlations among vegans prompts questions about the dynamics within this group, necessitating further research. The unexpected findings regarding vegetarians imply potential evolving preferences within this diet group, emphasizing the need for deeper investigations into the mechanisms driving such changes. In that respect, the study’s absence of specific plant-based options may constrain the scope of dietary analysis for individuals adhering to plant-based diets. While recognizing the growing popularity of plant-based alternatives, it is necessary to include diverse food options to represent various dietary preferences accurately.

In addition to the acknowledged limitations, in a broader sense, it is crucial to highlight the potential impact of socio–economic and cultural factors on dietary habits, which should be extensively explored. Variability in income, education, and cultural background can significantly influence food choices and preferences, introducing a layer of complexity not fully addressed in the current investigation. Furthermore, while providing a snapshot of dietary patterns at a specific time, the study’s cross-sectional design needs to capture the dynamic nature of individuals’ lifestyles and evolving dietary habits. Additionally, the study predominantly relies on quantitative data, limiting the exploration of qualitative aspects such as individual motivations, cultural influences, and the role of social environments in shaping dietary choices. Integrating qualitative methods could offer a nuanced understanding of the multifaceted aspects influencing individuals’ relationships with food. Addressing these limitations would contribute to a more holistic interpretation of the complex interplay between childhood and adult dietary habits.

## 5. Conclusions

The results of this study reveal several noteworthy patterns in dietary preferences across different diet groups, shedding light on the long-term impact of childhood food choices. Overall, these findings hint at the long-lasting impact of childhood dietary habits on adult food choices. This information can offer valuable insights to future research in order to develop nutritional interventions and public health initiatives that encourage healthier eating behaviors, considering diverse dietary preferences. These findings contribute valuable insights into individuals’ long-term dietary patterns and preferences based on their childhood diet, highlighting variations across diet groups and providing a practical framework for testing ToF. The cumulative evidence supports the notion that early dietary habits can have a lasting impact on adult food preferences, reinforcing the conclusions drawn in the hypothetical study provided. The current study presented practical tools for testing the behavioral manifestations of the “theory of food” (ToF), utilizing a tailored computerized paradigm. The findings reveal robust correlations between childhood and adult food choices among omnivores, varying patterns among vegetarians, and mixed results among vegans. Future studies should further assess the combination of the computerized food preferences paradigm with other food preference questionnaires to provide a robust and multifaceted approach, broadening data collection and enhancing understanding by triangulating self-reported information with real-time, interactive responses. Furthermore, the studies should also delve into the neuro-cognitive mechanisms underlying these behavioral manifestations.

## Figures and Tables

**Figure 1 nutrients-16-00428-f001:**
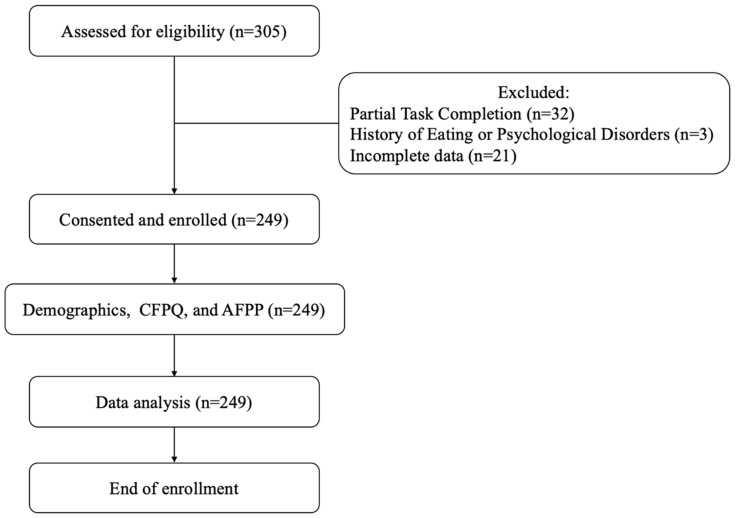
This flowchart outlines the participant enrollment process, including the criteria for inclusion/exclusion, the use of various tools and platforms, and the data analysis.

**Figure 2 nutrients-16-00428-f002:**
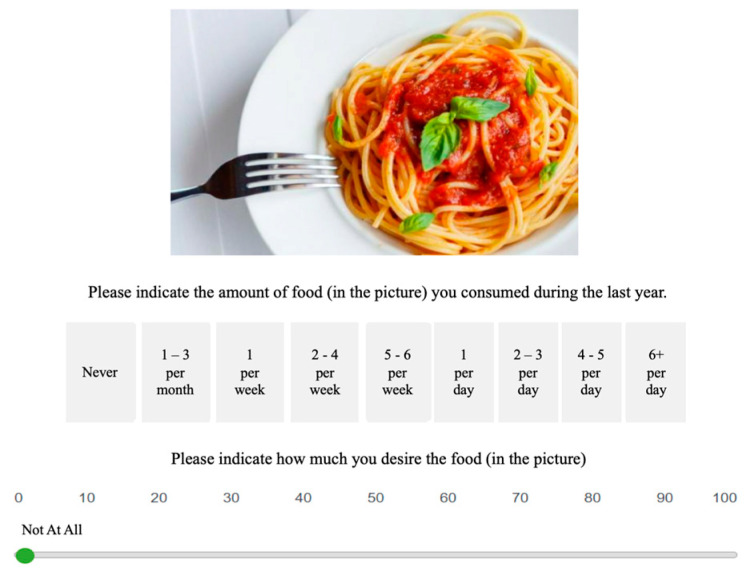
Depicts an example of the Adulthood Food Preferences Paradigm (AFPP). The AFPP included 76 colored images using commonly consumed dishes in Israel from seven significant categories: bread and cereals, fast food, dairy products, sweets and snacks, drinks, fruits and vegetables, and meat and fish. Participants were requested to state to which degree they consumed the dish over the past year on a Likert scale ranging from 1 = less than once a month to 9 = six times a day. Afterward, the participant was requested to note how much he desires this dish on an axis range from 0 = not craving at all to 100 = every craving.

**Figure 3 nutrients-16-00428-f003:**
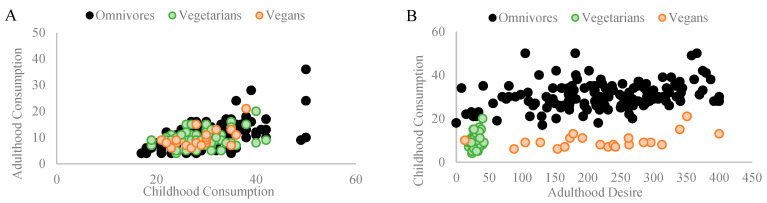
Illustrates significant Pearson coefficient correlations within different diet groups regarding the influence of childhood grain consumption on adult dietary patterns and preferences. (**A**) Among omnivores (n = 154), a robust correlation (rp = 0.502 and *p* = 0.001) was observed between grains consumed during childhood and adulthood. For vegetarians (n = 74), a correlation (rp = 0.28 and *p* = 0.016) between childhood and adult grain consumption was noted. Among vegans (n = 21), a substantial correlation (rp = 0.578 and *p* = 0.006) existed for grain consumption from childhood to adulthood. (**B**) Among omnivores, a significant correlation (rp = 0.299 and *p* = 0.001) emerged between childhood grain consumption and the desire for grains in adulthood. No correlations were found among vegetarians and vegans for the desire to consume grains (rp = 0.134, *p* = 0.256, and n.s.; rp = 0.053, *p* = 0.820, and n.s., respectively).

**Figure 4 nutrients-16-00428-f004:**
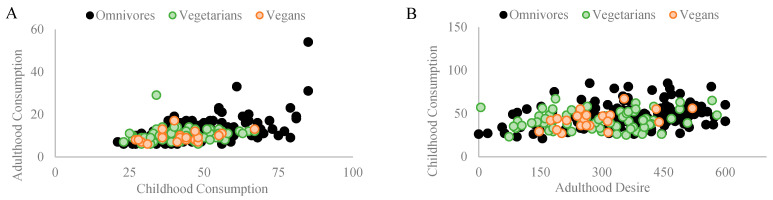
The figure demonstrates significant Pearson coefficient correlations in different diet groups regarding the impact of childhood fast food consumption on adult dietary patterns and preferences. (**A**) Among omnivores (n = 154), a notable correlation (rp = 0.527 and *p* = 0.001) exists between childhood and adult fast food consumption. For vegetarians (n = 74) and vegans (n = 21), no significant correlations were found between childhood fast food consumption and adult consumption (rp = 0.138, *p* = 0.240, and n.s.; rp = 0.351, *p* = 0.118, and n.s., respectively). (**B**) A correlation (rp = 0.276 and *p* = 0.001) between consuming fast food in childhood and the desire for fast food in adulthood was observed among omnivores. A similar correlation was found in vegans (rp = 0.518 and *p* = 0.016) but not among vegetarians (rp = 0.115, *p* = 0.325, and n.s.).

**Figure 5 nutrients-16-00428-f005:**
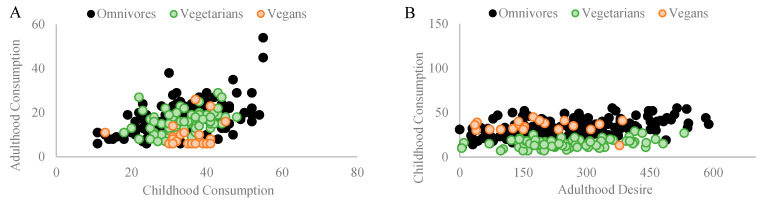
Highlights significant Pearson coefficient correlations in different diet groups concerning the consumption of dairy products from childhood to adulthood. (**A**) Among omnivores (n = 154), a strong correlation (rp = 0.538 and *p* = 0.001) is evident between childhood and adult dairy product consumption. For vegetarians (n = 74), a notable correlation (rp = 0.359 and *p* = 0.002) exists between childhood and adult dairy consumption. Among vegans (n = 21), no significant correlations are observed between childhood dairy consumption and adult consumption (rp = 0.159, *p* = 0.490, and n.s.). (**B**) A significant correlation (rp = 0.445 and *p* = 0.001) between childhood dairy consumption and the desire for it in adulthood is evident among omnivores. Among both vegetarians and vegans, no correlation exists between dairy consumption in childhood and the desire to consume it in adulthood (rp = −0.158, *p* = 0.493, and n.s.; rp = 0.173, *p* = 0.141, and n.s., respectively).

**Figure 6 nutrients-16-00428-f006:**
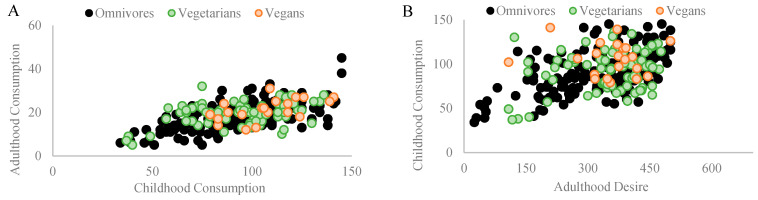
Illustrates significant Pearson coefficient correlations within different diet groups regarding childhood and adult consumption patterns of vegetables and fruits. (**A**) Among omnivores (n = 154), a robust correlation (rp = 0.642 and *p* = 0.001) is observed. For vegetarians (n = 74), a similar significant correlation exists between childhood and adult consumption (rp = 0.388 and *p* = 0.001). Among vegans (n = 21), no correlation was found with actual consumption of vegetables and fruits in adulthood (rp = −0.081, *p* = 0.726, and n.s.). (**B**) Among omnivores, vegetarians, and vegans, a significant correlation was observed between consuming vegetables and fruits in adulthood and the desire to consume them in adulthood (rp = 0.592 and *p* = 0.001; rp = 0.279 and *p* = 0.016; rp = 0.574 and *p* = 0.007, respectively).

**Figure 7 nutrients-16-00428-f007:**
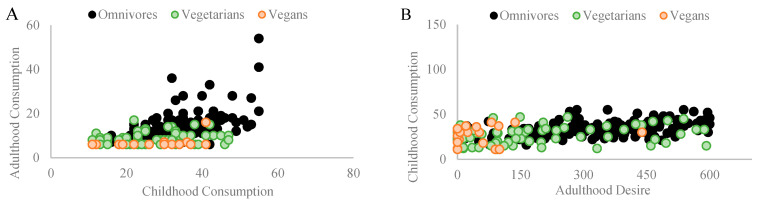
Depicts significant Pearson coefficient correlations within different diet groups regarding the influence of childhood meat consumption on adult dietary patterns and preferences. (**A**) Among omnivores (n = 154), a substantial correlation (rp = 0.528 and *p* = 0.001) exists between childhood and adult meat consumption. For vegetarians (n = 74) and vegans (n = 21), no correlation is found between childhood and adult meat consumption (rp = 0.317 and *p* = 0.597; rp = 0.33, *p* = 0.140, and n.s., respectively). (**B**) Among omnivores, a significant correlation was found between consuming meat in childhood and the desire to consume it in adulthood (rp = 0.415 and *p* = 0.001). Surprisingly, a similar correlation was found among vegetarians (rp = 0.381 and *p* = 0.001). No such correlation was observed in vegans (rp = 0.106, *p* = 0.646, and n.s.).

**Figure 8 nutrients-16-00428-f008:**
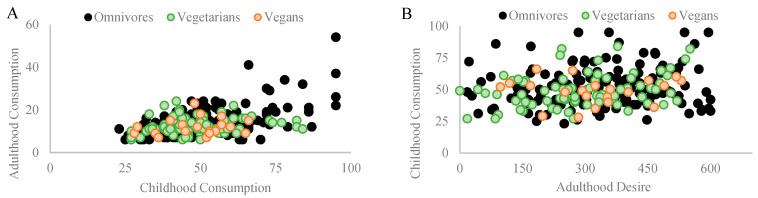
Illustrates significant Pearson coefficient correlations within different diet groups concerning the impact of childhood snack consumption on adult dietary patterns and preferences. (**A**) Among omnivores (n = 154), a robust correlation (rp = 0.567 and *p* = 0.001) exists between childhood and adult snack consumption. For vegetarians (n = 74), a marginal correlation was found between childhood and adult snack consumption (rp = 0.205 and *p* = 0.080). Among vegans (n = 21), no significant correlation was observed between childhood snack consumption and adult consumption (rp = 0.129, *p* = 0.578, and n.s.). (**B**) Among omnivores and vegetarians, a significant correlation was observed between childhood snack consumption and the desire to consume it in adulthood (rp = 0.271 and *p* = 0.001; rp = 0.329 and *p* = 0.004, respectively). no such correlation was found among vegans (rp = 0.034, *p* = 0.885, and n.s.).

**Table 1 nutrients-16-00428-t001:** Summarizes the study’s participant demographics, presenting the participant’s average age, STDEV (standard deviation), and range. It also presents gender distribution and insights into current and childhood living places, religious affiliations, health diet prevalence, health status distribution, and diet preferences.

	Mean	STDEV	Range		
*Age*	26.65	7.54	17–74		
	**Males**	**Females**			
*Gender*	86 (34.5%)	163 (65.5%)			

	**City**	**Kibbutz**	**Cooperative Settlement**	**Other**	
*Current living place*	98 (39.4%)	63 (25.3%)	65 (26.1%)	23 (9.2%)	
*Childhood living place*	127 (51%)	22 (8.8%)	50 (20.1%)	50 (20.1%)	

	**Muslim**	**Christian**	**Jew**	**Druze**	**Other**
*Religion*	10 (4%)	5 (2%)	219 (88%)	9 (3.6%)	6 (2.4%)

	**Yes**	**No**			
*Health Diet*	62 (24.9%)	187 (75.1%)			
	**Excellent**	**Good**	**Reasonable**	**Not so Good**	**Bad**
*Health Status*	99 (39.8%)	124 (49.8%)	22 (8.8%)	3 (1.2%)	1 (0.4%)

	**Omnivores**	**Vegetarian**	**Vegan**		
*Diet Preference*	154 (61.8%)	74 (29.8%)	21 (8.4%)		

## Data Availability

The datasets generated and/or analyzed during the current study are not publicly available due to their containing information that could compromise the privacy of research participants but are available from the corresponding author upon reasonable request.

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
