# Peer review of "Theory of Food: Unravelling the Lifelong Impact of Childhood Dietary Habits on Adult Food Preferences across Different Diet Groups"

_nutrients, 2024, doi:10.3390/nu16030428_

Round 1

Reviewer 1 Report

Comments and Suggestions for Authors

An impressive and novel piece of research that I found was well written. The concept of ToF is very interesting. 

My main concern is that of the methodology that requires greater explanation. 

1. How were participants recruited? Was there an exclusion/inclusion criteria?

2. The Childhood preferences questionnaire, it seems the authors developed the questionnaire. Is this correct. Has the questionnaire been validated elsewhere? Difficult to ask an adult what food groups they consumed as a child. Dietary self-report is notoriously bad, but added years makes it even harder. Can the authors justify the use of the questionnaire for this purpose?

3. Same comment regarding the Adult Food preferences. Did the authors develop the questionnaire? Is it validated. There are validated food preferences questionnaires available (like the Leeds FP questionnaire). Why not use that?

The food preferences categories appear to centre around meat, that is carnivore, vegetarian, vegan. It might be useful to outline why the researchers focused on these food preferences in this research? Perhaps comment on that in the introduction.

The discussion has been well written and interprets the findings appropriately. 

Author Response

Comment 1:

 How were participants recruited?

This is indeed a vital comment raised by the reviewer. A text was added on page 3, line 136:

“…(between December 2022 to February 2023) through a multifaceted approach, utilizing snowball sampling, word of mouth, and various social media platforms to ensure a diverse and representative sample.”

Comment 2:

Was there an exclusion/inclusion criteria?

This is a valuable comment by the reviewer. A text was added on page 3, lines 138-139:

“Participants were recruited based on age, generally healthy, not suffering from any chronic or mental disease, and not chronically medicated.”

Comment 3:

The Childhood preferences questionnaire, it seems the authors developed the questionnaire. Is this correct. Has the questionnaire been validated elsewhere?

Indeed, the CFPQ was developed in our lab. A description of the preliminary validation process was added to the text on page 4, lines 158-169, as follows:

“The CFPQ validation process was executed meticulously to ensure the reliability and credibility of the collected data. Content validation involved a comprehensive dietician review and survey methodology, confirming that the questionnaire effectively covered all relevant subject aspects. A pilot test was conducted with a representative sample (n=40) from the target population, revealing valuable insights into potential ambiguities or issues with question wording. The questionnaire underwent refinement for improved clarity and precision. Test-retest reliability measures were employed to assess the consistency of responses over time, further validating the questionnaire's stability. The culmination of these validation steps assures that the CFPQ is a valid instrument capable of accurately capturing information on the frequency of food consumption during the subject's childhood within the intended population.”

Comment 4:

Difficult to ask an adult what food groups they consumed as a child. Dietary self-report is notoriously bad, but added years makes it even harder. Can the authors justify the use of the questionnaire for this purpose?

We agree with the reviewer’s comment. However, despite inherent challenges associated with self-reporting, a questionnaire is valuable for gathering information on an adult's childhood food consumption. Questionnaires provide a structured and standardized approach, allowing participants to reflect on their past dietary habits at their own pace. While there may be limitations in recall accuracy, the questionnaire format helps capture a snapshot of participants' perceptions and memories, providing valuable insights into long-term dietary patterns that may contribute to a more comprehensive understanding of health and nutrition over the life course. Additionally, careful design and clear instructions enhance the reliability of self-reported data, making the questionnaire a practical and efficient method for exploring personal historical dietary behaviors. Nevertheless, in light of the reviewer’s comment, a text was added to the study’s limitations section on page 13, lines 424-429:

 “The use of the CFPQ  proves valuable in eliciting information on an adult's childhood food consumption by offering a structured and standardized approach, allowing participants to reflect at their own pace; despite potential recall limitations, the format captures valuable insights into long-term dietary patterns, contributing to a comprehensive understanding of health and nutrition, with careful design and clear instructions enhancing the reliability of self-reported data.”

Comment 5:

Same comment regarding the Adult Food preferences. Did the authors develop the questionnaire? Is it validated.

the AFPP was developed in our lab. A description of the preliminary validation process was added to the text on page 5, lines 182-193, as follows:

“In the validation process of AFPP, rigorous measures were implemented to ensure the reliability and accuracy of the collected data. Firstly, the task underwent expert review by nutritionists and psychologists to ascertain its alignment with established dietary guidelines and psychological constructs related to eating behavior. Subsequently, a pilot study (n=17) was conducted with diverse participants to identify potential ambiguities or misconceptions in task instructions. Adjustments were made based on the feedback received, and the finalized task was then administered to a representative sample of the target population (n=35). Participants' responses were cross-verified during data collection with traditional dietary assessments, such as food diaries and 24-hour recalls, to validate the computerized task's outcomes. Statistical analyses, including correlation and reliability assessments, were employed to establish the task's consistency and validity in measuring food consumption patterns in adulthood.”

Comment 6:

There are validated food preferences questionnaires available (like the Leeds FP questionnaire). Why not use that?

This is an interesting point raised by the reviewer. The computerized paradigm used in the current study introduces a dynamic and interactive element, utilizing visual stimuli, multimedia, or interactive tasks to elicit more nuanced responses than traditional paper-based methods. This approach allows for a more immersive and engaging exploration of participants' preferences. On the other end, we acknowledge that the Leeds FP questionnaire provides a structured framework for obtaining self-reported data on food preferences. However, since we find the reviewer’s comment necessary, we added the following text on page 14, lines 467-471 at the end of the discussion concerning future directions:

“Future studies should further assess the combination of the computerized food preferences paradigm with other food preference questionnaires to provide a robust and multifaceted approach, broadening data collection and enhancing understanding by triangulating self-reported information with real-time, interactive responses”.

Comment 7:

The food preferences categories appear to centre around meat, that is carnivore, vegetarian, vegan. It might be useful to outline why the researchers focused on these food preferences in this research? Perhaps comment on that in the introduction.

We thank the reviewer for this comment. A text was added to the introduction on page 3, lines 120-127:

“Specifically, by studying omnivores, vegetarian, and vegan food preferences, recognizing their profound impact on health, environmental sustainability, and ethical considerations. By focusing on these categories, our research aims to delve into the multifaceted implications of diverse diets, offering valuable insights into the intersection of personal choices, societal trends, and overall well-being. This emphasis aligns with the increasing importance of comprehending varied dietary patterns amid contemporary health and environmental sustainability challenges [33]

33 - Nelson ME, Hamm MW, Hu FB, et al. Alignment of Healthy Dietary Patterns and Environmental Sustainability: A Systematic Review. Advances in Nutrition 2016; 7: 1005–1025.

Comment 8:

The discussion has been well written and interprets the findings appropriately.

We thank the reviewer for this supporting comment. 

Reviewer 2 Report

Comments and Suggestions for Authors

Dear Editors,

Dear Authors,

I am happy to help improving further the quality of your manucript based on my subsequent comments;

thank you for the chance to review and contribute to help you improving this very interesting, important and future-oriented topic and promising title, of that I am convinced - but not after a focused and well-ealborated MAJOR revision.

The title above all is promising but the subsequet text could not hold the promise of title and fails to shed light to the background and also leaves an interested reader like me with more unanswered questions than feeling happy with the read; however, I d highly recommend to make consult a scientific supervisor along with elaborating my detailed comments below, will results hopefully and eventually into a nice paper out of this failed first version, although I feel the topic has potential left to be tapped and uncovered - issue left to do!

To summarize my main concerns: many unanswered questions and poor structure - issue to fix poor academic writing, and adding meaningful literture to the reference list. 

After a deep re-thinking and focused major revision adressing and nicely elaborating the comments of all reviewers it might be ready being considered for next round of review.

Best and all good success! 

xxx

General

Terminology, starting with Abstract, line 17: I´d highly recommend the terminology of carnivore as seems totally wrong here and must be replaced by mixed diet (maybe heavy centered on meat, ok) - but carnivores are from the feline species group such as tiger, cat, etc. with body architecture totally different to herbivores, frugivores - and so is another group/type of definition by anatomy/physiology/body architecture rather than focusing on dietary patterns - be careful with such wordings!

in-text citation: Allen (YEAR is missing) - over pages 1-2, not clear if always ref. no. 1 is referred to or other, Allen down to line 71 - mainly on him!! is insufficient

Academic writing is a big issue with this paper and must be fixed!

Supplement

CFPQ needs a reference, also as supplmemental material

for all the animal-sourced products/foods etc. - how did the vegans and vegetarians know that naturally for them it was meant the plant-based or plant-only "vish" of "plant-meat" or "plant-milk" version of what you display/ask for in a specific item?

Abstract 

Terminology (line 17)

correlations between which variables/considering what? ... in dietary subgroups?

Conclusion is not possible to base on the results presented, as results are not clear, so conclusion "hanging loose" - need to display hard facts as results

Terminology (line 37)

human vs. non-human primates - why do we need this since the title is talking about children and adults/humans?

References

-> some key papers are lacking, eg. this list, but not limited to, no claim of comprehensiveness - must be extended by meaningful key/landmark literature directly related to the lifecourse/track from childhood to adulthood

x Agirbasli M, Tanrikulu AM, Berenson GS. Metabolic Syndrome: Bridging the Gap from Childhood to Adulthood. Cardiovasc Ther. 2016 Feb;34(1):30-6. doi: 10.1111/1755-5922.12165.

x Delaney L, Smith JP. Childhood health: trends and consequences over the life course. Future Child. 2012 Spring;22(1):43-63. doi: 10.1353/foc.2012.0003.

x Sandrine Lioret 1,*, Karen J. Campbell 2, Sarah A. McNaughton 2, Adrian J. Cameron 3, Jo Salmon 2, Gavin Abbott 2 and Kylie D. Hesketh 2Lifestyle Patterns Begin in Early Childhood, Persist and Are Socioeconomically Patterned, Confirming the Importance of Early Life Interventions. Nutrients 2020, 12, 724; doi:10.3390/nu12030724

x AND 2016 Position paper on vegetarian diets - lacking; only 2009

x Boštjan Jakše 1, Zlatko Fras 2 3, Nataša Fidler Mis 4 5. Vegan Diets for Children: A Narrative Review of Position Papers Published by Relevant Associations. Nutrients 2023 Nov 7;15(22):4715. doi: 10.3390/nu15224715.

... with only 1 = no. 1 Allen 2012 directly linked to title and theme of ToF! - with 2012 does not support your statement " .. novel theoreatical framework ...." - seems too less and too old to strengthen your opionon!

Figures

nice figures, maybe coloured would add to the paper - but do they add meaningfully this way as most/lots of/all dots are very very close (except of 2)??

is Fig. 7 for Soft drinks really needed - seems a bit a sharp contrast to all other food-related topics/groups, why not a small paragraph only, and so stay with a stringent structure - here it does not match (if you want to keep, after snacks) 

i miss a flow chart for participant enrollment (compliance for exclsuion-inclusion)

Tables

legends must be above table and must be self-explaining!

Table 1 - what is STDEV - is not defined

Table 1 is confusing, hard to read - and intersted readeris loosing interst - what I would regret!

xxx

Specific

Intro (approx. 1 1/2 pages)

1st sentence (1st paragraph - must be last at intro!) directly AIM is much too early - is not clear what the gap in literature is, not clear what the background to issue is, not clear what the research question is - issue to academic writing! <-> do not explain/display what you will do in the following

line 58 - eating itself - what does this mean?

half a page TOM, primates, chimapnzees - etc. why is this necessary - it does not help further when an intested reader wants to learn about what is promised in the text

line 72 - many studies is more than 4 papers (5-8)

lines 92-92: ToF recently developed - seems incorrect with the reference from 2012!

Again at the end of Intro: AIM - but still we do not know the gap, only what the author did investigate

Last sentence here is practical implication and has to be removed to Conslusion or Disc. not here!

to sum up: this introduction is not shedding light on the relevant background, does not sell what is maybe intersting to come after in methods, results etc.

Method (approx. 2 pages)- is lacking but is key to any review, pls add meaninfully, thx

line 104: how did you recruit, where, within which time-frame, what are the exclusion vs. inclusion criteria - ? all unclear

line 105: experiment - what experiment? do you have an ethic approval of an ethics board; exclusion of N=56 is insufficient, has to be split by any aspect seperately, not pooled -> flow chart!

After 2 lines end of method - is insufficient

then Stats from line 106 - belongs to very end of methods chapter

Terminology and definition of dietary subgroups - how did you assign them to: random, volunatary - did you control for diet type, if yes, how did you do?

many unanswered questions and poor structure

lines 117: it was an experiment or a questionnaire (I understand the CFPQ and AFPP, but not the overall presentation to participants!) or a dietary recall or an interview - what was it?

was it online or face-to-face?

how can a conclusion been drawn if not both a subpopulation of kids and adults were studied, at best the same sample 20 years later - ?? is it a cross-sect. or a longitudinal (at best intra-subject follow-up)? all not clear to me

Again at the end: Statistics subchapter

Results (approx. 5 1/2 pages) - I quit reading after Methods

Resutls start with statistics prodecure = no go!

Discussion (approx. 1 1/2 pages) - I quit reading after Methods

is too short to discuss the big amount of results and also in relation to Intro (if Into is 1 page, Disc must be about 2-3 pages -> ratio 1 : 2-3)

1st sentence, 1st line - "... we investigated" - who is we if you are 1 author only?

lines 345-355: Guess limitations are to be extended ...

line 356: "yes again ..." is not good to start closing the discussion

line 358: guess this conslusion cannot be drawn based on this method

Conclusion/Take Home Message - I quit reading after Methods

to me there is no take home message, only what future studies on ToF shall delve into - far too early after me!

Author Response

Reviewer #2

Comment 1:

Terminology, starting with Abstract, line 17: I´d highly recommend the terminology of carnivore as seems totally wrong here and must be replaced by mixed diet (maybe heavy centered on meat, ok) - but carnivores are from the feline species group such as tiger, cat, etc. with body architecture totally different to herbivores, frugivores - and so is another group/type of definition by anatomy/physiology/body architecture rather than focusing on dietary patterns - be careful with such wordings!

We agree and apologize for using the term “carnivores.” Although the term is often used to describe human eating habits, unlike obligate carnivores like cats, humans are not exclusively dependent on meat for their nutritional needs. They can derive essential nutrients from a diverse range of food sources. Therefore, the term "omnivore" more accurately characterizes the dietary habits of humans. Therefore, the term was changed throughout the manuscript where needed, and the graphs were edited accordingly. 

Comment 2:

in-text citation: Allen (YEAR is missing) - over pages 1-2, not clear if always ref. no. 1 is referred to or other, Allen down to line 71 - mainly on him!! is insufficient

Academic writing is a big issue with this paper and must be fixed!

We thank the reviewer for these comments. The “theory of food” idea is novel. Unfortunately, it did not receive the appropriate attributes in the scientific literature. In fact, this is part of the reasons that led to the current paper. Nevertheless, we acknowledge the importance of broadening the scientific references concerning the neurocognitive bases of human multimodal food perception. Therefore, a text was added to the introduction on page 3, lines 106-112:

“Understanding intraoral food awareness in humans involves the concurrent and sustained activation of specific neuronal subsets related to affect and hedonics, those associated with food identity, and those contributing to affect and identity [31]. In light of the current study, particular multifaceted interactions may involve correspondence synesthesia, suggesting intriguing avenues for exploration of the effects of early-life food-related sensory experiences on adulthood learning [32]. This calls for further examination of the anatomical and functional developmental connections among brain structures crucial for understanding correspondence synesthesia and food preference”

 31 - Verhagen JV. The neurocognitive bases of human multimodal food perception: Consciousness. Brain Research Reviews 2007; 53: 271–286.

32 - Verhagen JV, Engelen L. The neurocognitive bases of human multimodal food perception: Sensory integration. Neuroscience & Biobehavioral Reviews 2006; 30: 613–650.

 In addition, several more papers were incorporated into the MS. Page 2, line 57:

6 - Enriquez JP, Archila-Godinez JC. Social and cultural influences on food choices: A review. Critical Reviews in Food Science and Nutrition 2022; 62: 3698–3704.

7 - Spencer SJ, Korosi A, Layé S, et al. Food for thought: how nutrition impacts cognition and emotion. npj Sci Food 2017; 1: 7.

8 - Birch LL. DEVELOPMENT OF FOOD PREFERENCES. Annu Rev Nutr 1999; 19: 41–62.

Comment 3:

CFPQ needs a reference, also as supplmemental material.

The CFPQ was developed in our lab. A description of the preliminary validation process was added to the text on page 4, lines 158-169, as follows:

“The CFPQ validation process was executed meticulously to ensure the reliability and credibility of the collected data. Content validation involved a comprehensive dietician review and survey methodology, confirming that the questionnaire effectively covered all relevant subject aspects. A pilot test was conducted with a representative sample (n=40) from the target population, revealing valuable insights into potential ambiguities or issues with question wording. The questionnaire underwent refinement for improved clarity and precision. Test-retest reliability measures were employed to assess the consistency of responses over time, further validating the questionnaire's stability. The culmination of these validation steps assures that the CFPQ is a valid instrument capable of accurately capturing information on the frequency of food consumption during the subject's childhood within the intended population.”

Comment 4:

for all the animal-sourced products/foods etc. - how did the vegans and vegetarians know that naturally for them it was meant the plant-based or plant-only "vish" of "plant-meat" or "plant-milk" version of what you display/ask for in a specific item?

This is an insightful comment raised by the reviewer. In the current study, items did not explicitly include plant-based or plant-only alternatives like "vish," "plant-meat," or "plant-milk." This was clear to the participants when given the food categories in the questionnaire/task instructions. Nevertheless, the absence of specific plant-based options in the study might limit the scope of dietary analysis for these individuals. Given the increasing popularity of plant-based alternatives, we acknowledge the essential inclusion of diverse food options to capture the nuances of various dietary preferences more accurately. Thus, the following text was added on page 14, lines 435-439:

“In that respect, the study's absence of specific plant-based options may constrain the scope of dietary analysis for individuals adhering to plant-based diets. While recognizing the growing popularity of plant-based alternatives, it is necessary to include diverse food options to represent various dietary preferences accurately.”

Comment 5:

Terminology (line 17)

correlations between which variables/considering what? ... in dietary subgroups?

We agree with the reviewer’s comment. Indeed, not clear enough, a text was added to the abstract on page 1, lines 19-22:

“Notably, omnivores show correlations in grains, fast food, dairy products, vegetables, meat, soft drinks, and snack consumption. Vegetarians exhibit correlations in grains, fast food, dairy products, vegetables, snacks, and, surprisingly, meat consumption. Vegans display correlations in grains, fast food, vegetables, and snacks.”

Comment 6:

Conclusion is not possible to base on the results presented, as results are not clear, so conclusion "hanging loose" - need to display hard facts as results

We accept the reviewer’s comment. We toned down the language and the concluding remark in the abstract was changed on page 1, lines 22-24:

“The study suggests that childhood dietary habits tend to influence adult food choices, offering insights for future research in the field of Theory of Food (ToF).”

Comment 7:

Terminology (line 37)

human vs. non-human primates - why do we need this since the title is talking about children and adults/humans?

This is indeed true. The attribution to chimpanzees is irrelevant for the purpose of the current study. Therefore, the text was modified on page 2, lines 46-48:

“The research proposes that humans possess a unique intelligence focused on understanding mental states, suggesting human cognitive specialization for the TOM [4].”

4 - Povinelli DJ, Preuss TM. Theory of mind: evolutionary history of a cognitive specialization. Trends in Neurosciences 1995; 18: 418–424.

 Comment 8:

-> some key papers are lacking, eg. this list, but not limited to, no claim of comprehensiveness - must be extended by meaningful key/landmark literature directly related to the lifecourse/track from childhood to adulthood

x Agirbasli M, Tanrikulu AM, Berenson GS. Metabolic Syndrome: Bridging the Gap from Childhood to Adulthood. Cardiovasc Ther. 2016 Feb;34(1):30-6. doi: 10.1111/1755-5922.12165.

x Delaney L, Smith JP. Childhood health: trends and consequences over the life course. Future Child. 2012 Spring;22(1):43-63. doi: 10.1353/foc.2012.0003.

x Sandrine Lioret 1,*, Karen J. Campbell 2, Sarah A. McNaughton 2, Adrian J. Cameron 3, Jo Salmon 2, Gavin Abbott 2 and Kylie D. Hesketh 2Lifestyle Patterns Begin in Early Childhood, Persist and Are Socioeconomically Patterned, Confirming the Importance of Early Life Interventions. Nutrients 2020, 12, 724; doi:10.3390/nu12030724

x AND 2016 Position paper on vegetarian diets - lacking; only 2009

x Boštjan Jakše 1, Zlatko Fras 2 3, Nataša Fidler Mis 4 5. Vegan Diets for Children: A Narrative Review of Position Papers Published by Relevant Associations. Nutrients 2023 Nov 7;15(22):4715. doi: 10.3390/nu15224715.

We highly thank the reviewer for turning our attention to this issue and for suggesting the above papers. In light of the reviewer’s comment, the following text was added to the introduction, specifically concerning the life course from childhood to adulthood with a focus on food and nutrition on page 2, lines 75-85:

Understanding the life-course trajectory from childhood to adulthood concerning food is crucial for promoting lifelong health and well-being. Various studies investigated different aspects of dietary patterns and their implications over time. For instance, it was reported that food intake aligns with recommendations; adolescent habits moderately predict adult intake, influenced by gender, location, and socioeconomic status [9]. Longitudinal changes in diet and the tracking of fruit and vegetable consumption uncovered links to cardiovascular health and overweight [10]. Further, exploring dietary patterns and their associations with cardiometabolic risk factors across the lifespan [11–13]. These investigations underscore the need for a comprehensive understanding of dietary trajectories, advocating for informed interventions that can shape nutritional habits from early childhood through adulthood [14].”

9 - Lake AA, Mathers JC, Rugg-Gunn AJ, et al. Longitudinal change in food habits between adolescence (11–12 years) and adulthood (32–33 years): the ASH30 Study. J Public Health 2006; 28: 10–16.

10 - Te Velde SJ, Twisk JWR, Brug J. Tracking of fruit and vegetable consumption from adolescence into adulthood and its longitudinal association with overweight. Br J Nutr 2007; 98: 431–438.

11 - Agirbasli M, Tanrikulu AM, Berenson GS. Metabolic Syndrome: Bridging the Gap from Childhood to Adulthood. Cardiovasc Ther 2016; 34: 30–36.

12 - Herrmann SD, Angadi SS. Children’s Physical Activity and Sedentary Time and Cardiometabolic Risk Factors. Clin J Sport Med 2013; 23: 408–409.

13 - Mishra GD, McNaughton SA, Bramwell GD, et al. Longitudinal changes in dietary patterns during adult life. Br J Nutr 2006; 96: 735–744.

14 - Delaney L, Smith JP. Childhood Health: Trends and Consequences over the Life Course. Future Child 2012; 22: 43–63.

Comment 9:

... with only 1 = no. 1 Allen 2012 directly linked to title and theme of ToF! - with 2012 does not support your statement " .. novel theoreatical framework ...." - seems too less and too old to strengthen your opionon!

We accept the reviewer’s comment. Concerning this and the rest of the reviewer’s comments, the MS was modified, the language was toned where needed, and additional scientific supports were added.

Comment 10:

nice figures, maybe coloured would add to the paper - but do they add meaningfully this way as most/lots of/all dots are very very close (except of 2)??

We thank the reviewer for this comment. Colors were added to the graphs.

Comment 11:

is Fig. 7 for Soft drinks really needed - seems a bit a sharp contrast to all other food-related topics/groups, why not a small paragraph only, and so stay with a stringent structure - here it does not match (if you want to keep, after snacks).

This is a valuable comment raised by the reviewers. The text was rearranged, and the Snacks results now appear before the results of the drink. Furthermore, as suggested by the reviewer, figure 7 (soft drinks) was removed from the MS. Instead, a text was added on page 12, lines 343-345:

“While the findings on food emphasize nutritional aspects and dietary recommendations, the exploration of drinks delves into beverage consumption patterns and their distinct impact on overall health. Thus, the data is not presented”  

Comment 12:

i miss a flow chart for participant enrollment (compliance for exclsuion-inclusion)

We thank the reviewer for this comment. Enrollment flow chart was added as Figure 1 on page 4:

Figure 1: This flowchart outlines the process of participant enrollment, including the criteria for inclusion/exclusion, the use of various tools and platforms, and the data analysis. 

 Comment 13:

legends must be above table and must be self-explaining!

We thank the reviewer for this comment; the text was changed accordingly. The table’s legend was re-written as follows (Page 6):

Table 1 summarizes the study's participant demographics, presenting the participant's average age, STDEV (standard deviation), and range. It also presents gender distribution and insights into current and childhood living places, religious affiliations, health diet prevalence, health status distribution, and diet preferences”

Comment 14:

Table 1 - what is STDEV - is not defined

The text was corrected; please see the previous comment response.

Comment 15:

Table 1 is confusing, hard to read - and intersted readeris loosing interst - what I would regret!

We thank the reviewer for turning our attention to this. The table was re-arranged to ease its reading (Pages 6-7).

Comment 16:

1st sentence (1st paragraph - must be last at intro!) directly AIM is much too early - is not clear what the gap in literature is, not clear what the background to issue is, not clear what the research question is - issue to academic writing! <-> do not explain/display what you will do in the following

We accept the reviewer’s comment. Yet, this is a matter of an academic writing style that might differ across continents or journals. However, we think that the reviewer is correct and that a better opening might better fit; thus, the text was modified on page 1, lines 28-36:

“The current study investigates the developing concept of the "Theory of Food" (ToF), aiming to analyze behavioral expressions related to individuals' core perceptions of food. With a broad exploration encompassing diverse dietary patterns, including those of omnivores, vegetarians, and vegans, the research acknowledges the significant impact of various diets on health, environmental sustainability, and ethical considerations. The study seeks to uncover the nuanced implications of personal choices. Examining participants' prospective food-related self-reported preference and a computerized food-related preference task aimed to set a groundwork for future investigations into the factors influencing cognitive food preference.”

Comment 17:

line 58 - eating itself - what does this mean?

We thank the reviewer for noticing this. To explain better the meaning of “eating itself”, a text was added on page 2, lines 60-61:

“…(i.e., the mental images, ideas, or cognitive associations linked to the act of eating)…

Comment 18:

half a page TOM, primates, chimapnzees - etc. why is this necessary - it does not help further when an intested reader wants to learn about what is promised in the text

This is indeed true. The text was shortened and modified on page 2, lines 49-54:

“In the intricate social dynamics of primates, adept Theory of Mind (TOM) abilities prove crucial. In TOM tasks, individuals observe a scenario where a character (X) places an object in a basket, exits, and is replaced by character Y. Subjects then report where X thinks the object is. Proficient TOM individuals comprehend others' perspectives, demonstrating a capacity to navigate the subjective world, interpret images, and engage in cognitive manipulations [5]

In addition, the attribution to chimpanzees was removed from the text to focus mainly on human findings. For example, see page 2, lines 46-48:

“The research proposes that humans possess a unique intelligence focused on understanding mental states, suggesting human cognitive specialization for the TOM [4].”

Comment 19:

line 72 - many studies is more than 4 papers (5-8)

We agree with the reviewer’s comment. The text was changed on page 2, lines 86:

 “Studies link the individual's nutritional choice to mood, physical health, and subjective mental well-being [15–18].”

Comment 20:

lines 92-92: ToF recently developed - seems incorrect with the reference from 2012!

Totally agree with the reviewer. The text was changed on page 1, line 37:

“The ‘theory of food' (ToF) yet to be studied in depth [1].”

Comment 21:

Again at the end of Intro: AIM - but still we do not know the gap, only what the author did investigate

This is a valuable comment raised by the reviewer. The following text was added on page 3, lines 114-119:

“Overall, research on ToF remains a relatively unexplored domain. While acknowledging the profound impact of diverse diets on health, environmental sustainability, and ethical considerations, there is a need for an in-depth examination of the implications of personal choices of food perception and their underlying neurocognitive mechanism. The current study aimed to reveal insights into some factors influencing food preferences and lay the groundwork for future research in this underexplored field.”  

Comment: 22:

Last sentence here is practical implication and has to be removed to Conslusion or Disc. not here!

We thank the reviewer for this comment and the text was removed.

Comment 23:

to sum up: this introduction is not shedding light on the relevant background, does not sell what is maybe intersting to come after in methods, results etc.

We hope that the reviewer will find the new version of the MS and, specifically, the introduction better. 

Comment 24:

Method (approx. 2 pages)- is lacking but is key to any review, pls add meaninfully, thx

We agree with the reviewer. The whole methods section was re-written, and text and tables were re-arranged and added; please see pages 3-6, lines 135-230.

Comment 25:

line 104: how did you recruit, where, within which time-frame, what are the exclusion vs. inclusion criteria - ? all unclear

This is a vital comment raised by the reviewer. A text was added on page 3, lines 135-140:

“…retrospective cross-sectional study (between December 2022 to February 2023) through a multifaceted approach, utilizing snowball sampling, word of mouth, and various social media platforms to ensure a diverse and representative sample. Participants were recruited based on age, generally healthy, not suffering from any chronic or mental disease, and not chronically medicated..”

Comment 26:

line 105: experiment - what experiment? do you have an ethic approval of an ethics board;

Yes, this is stated on page 3, lines 147-149:

“The ethics committee of THAC approved all procedures (Ethics #: 28-12/2022)”

Comment 27:

exclusion of N=56 is insufficient, has to be split by any aspect seperately, not pooled -> flow chart!

This is a meaningful comment. We included an enrolment flow chart to clarify this issue further, page 4.

Comment 28:

After 2 lines end of method - is insufficient then Stats from line 106 - belongs to very end of methods chapter

Sorry, not sure if understood correctly. If the new version of the methods section is unclear, we ask the reviewer to be more specific in this comment.

Comment 29:

Terminology and definition of dietary subgroups - how did you assign them to: random, volunatary - did you control for diet type, if yes, how did you do?

We thank the reviewer for this comment. The division of participants into the different diet groups was voluntary based on their self-reports during the enrollment to the study. This was added to text on page 3, lines 144-146:

“Participants voluntarily self-reported their dietary preferences during enrollment, leading to the categorization into distinct diet groups for the study:…”

Comment 30:

lines 117: it was an experiment or a questionnaire (I understand the CFPQ and AFPP, but not the overall presentation to participants!) or a dietary recall or an interview - what was it?

As described in the method section, the CFPQ is a questionnaire, while the AFPP is a computerized food-related task. For a complete description, please see pages 4-5, lines 156-178, and page 5, lines 180-209.

Comment 31:

was it online or face-to-face?

This is stated in the experimental procedure section on page 6, lines 211-214. To better clarify this issue, we have added additional text on page 6, line 213:

“…an online version…”

Comment 32:

how can a conclusion been drawn if not both a subpopulation of kids and adults were studied, at best the same sample 20 years later - ?? is it a cross-sect. or a longitudinal (at best intra-subject follow-up)? all not clear to me

This is a vital comment raised by the reviewer. The current study is a retrospective cross-sectional study, delving into the participant’s self-reports on their preference as adults and children. A text highlighting this was added on page 3, lines 135-136:

“…retrospective cross-sectional study…”

Comment 33: 

Results (approx. 5 1/2 pages) - I quit reading after Methods

Resutls start with statistics prodecure = no go!

The preliminary description for controlling for possible cofounds was removed from the results section and moved to the data analysis section on page 6, lines 226-230.

Comment 34: 

Discussion (approx. 1 1/2 pages) - I quit reading after Methods

is too short to discuss the big amount of results and also in relation to Intro (if Into is 1 page, Disc must be about 2-3 pages -> ratio 1 : 2-3)

We thank the reviewer for this comment. Changes were conducted across the whole MS, including the discussion. Specific text sections were incorporated throughout the discussion. For example, when dealing with the finding that vegetarians report a desire for meat, page 13, lines 383-394:

“The current study's revelation that vegetarians report a desire for meat holds substantial significance within the broader context of understanding dietary preferences and biases. This observation serves as compelling evidence for establishing food biases early in life, despite subsequent changes in attitudes toward meat during later stages of development [48]. It underscores the persistent influence of early experiences and exposures on individuals' food preferences, challenging the assumption that attitudes toward a specific food category, such as meat, are exclusively shaped by later-life factors [49]. The reported desire for meat among vegetarians may suggest a deep-seated inclination formed in the early stages of life, shedding light on the complexity of human food choices and emphasizing the need for comprehensive research to unravel the intricate interplay of psychological, cultural, and environmental factors in shaping dietary preferences across the lifespan in longitudinal designs.”

48 - Modlinska K, Pisula W. Selected Psychological Aspects of Meat Consumption—A Short Review. Nutrients 2018; 10: 1301.

49 - Çoker EN, Van Der Linden S. Fleshing out the theory of planned of behavior: Meat consumption as an environmentally significant behavior. Curr Psychol 2022; 41: 681–690.

In addition, when dealing with the correlations found for soft drink consumption between childhood and adulthood among omnivores and vegetarians and, in contrast, the lack of correlations among vegans regarding the desire to consume specific food items in adulthood based on childhood consumption. The following text was added on page 13, lines 410-418:

“This intriguing finding suggests a potential divergence in the impact of early dietary habits on adult preferences within the vegan community. The unique nature of the vegan diet [54], characterized by distinct nutritional choices and ethical considerations [55], may contribute to this differentiation. Further investigation into the factors shaping dietary continuity or change among vegans is warranted to deepen our understanding of the intricate interplay between early dietary experiences and adult food preferences within this population. This knowledge could inform tailored dietary interventions and enhance our appreciation of the complex dynamics influencing lifelong dietary choices among diverse dietary groups.”

54 - Gallagher CT, Hanley P, Lane KE. Pattern analysis of vegan eating reveals healthy and unhealthy patterns within the vegan diet. Public Health Nutr 2022; 25: 1310–1320.

55 - Beck V, Ladwig B. Ethical consumerism: Veganism. WIREs Climate Change 2021; 12: e689.

Last, we have extended the limitations section to address facets beyond the current study’s scope and address a broader perspective of studying food. Please see page 14, lines 440-452:

“In addition to the acknowledged limitations, in a broader sense, it is crucial to highlight the potential impact of socio-economic and cultural factors on dietary habits, which should be extensively explored. Variability in income, education, and cultural background can significantly influence food choices and preferences, introducing a layer of complexity not fully addressed in the current investigation. Furthermore, while providing a snapshot of dietary patterns at a specific time, the study's cross-sectional design needs to capture the dynamic nature of individuals' lifestyles and evolving dietary habits. Additionally, the study predominantly relies on quantitative data, limiting the exploration of qualitative aspects such as individual motivations, cultural influences, and the role of social environments in shaping dietary choices. Integrating qualitative methods could offer a nuanced understanding of the multifaceted aspects influencing individuals' relationships with food. Addressing these limitations would contribute to a more holistic interpretation of the complex interplay between childhood and adult dietary habits.”

Comment 35:

1st sentence, 1st line - "... we investigated" - who is we if you are 1 author only?

There are reasons for writing in the third person. First, writing a scientific paper in the first person is uncommon in most scientific disciplines, as the passive voice is traditionally preferred for objectivity. Second, as presented in the acknowledgments section, this work was conducted as a group in the physiology and behavior laboratory.   

Comment 36:

lines 345-355: Guess limitations are to be extended ...

Indeed, this section was extended on page 13, lines 424-429:

“…the use of the CFPQ  proves valuable in eliciting information on an adult's childhood food consumption by offering a structured and standardized approach, allowing participants to reflect at their own pace; despite potential recall limitations, the format captures valuable insights into long-term dietary patterns, contributing to a comprehensive understanding of health and nutrition, with careful design and clear instructions enhancing the reliability of self-reported data.”

On page 14, lines 435-439:

“In that respect, the study's absence of specific plant-based options may constrain the scope of dietary analysis for individuals adhering to plant-based diets. While recognizing the growing popularity of plant-based alternatives, it is necessary to include diverse food options to represent various dietary preferences accurately.”

And on page 14, lines 440-452:

“In addition to the acknowledged limitations, in a broader sense, it is crucial to highlight the potential impact of socio-economic and cultural factors on dietary habits, which should be extensively explored. Variability in income, education, and cultural background can significantly influence food choices and preferences, introducing a layer of complexity not fully addressed in the current investigation. Furthermore, while providing a snapshot of dietary patterns at a specific time, the study's cross-sectional design needs to capture the dynamic nature of individuals' lifestyles and evolving dietary habits. Additionally, the study predominantly relies on quantitative data, limiting the exploration of qualitative aspects such as individual motivations, cultural influences, and the role of social environments in shaping dietary choices. Integrating qualitative methods could offer a nuanced understanding of the multifaceted aspects influencing individuals' relationships with food. Addressing these limitations would contribute to a more holistic interpretation of the complex interplay between childhood and adult dietary habits.”

Comment 37:

line 356: "yes again ..." is not good to start closing the discussion

We thank the reviewer for this comment. The term was removed.

Comment 38:

line 358: guess this conslusion cannot be drawn based on this method

This is indeed true; we thank the reviewer. The language was toned down, and the text was modified on page 14, lines 455-458:

“These findings hint at the long-lasting impact of childhood dietary habits on adult food choices. This information can offer valuable insights to future research to develop nutritional interventions and public health initiatives that encourage healthier eating behaviors, considering diverse dietary preferences.”

Comment 39:

Conclusion/Take Home Message - I quit reading after Methods

to me there is no take home message, only what future studies on ToF shall delve into - far too early after me!

Hope you will find the revised version better.